# 3D fluorescence microscopy data synthesis for segmentation and benchmarking

**Dennis Eschweiler** [ID]*, **Malte Rethwisch, Mareike Jarchow, Simon Koppers** [ID], **Johannes Stegmaier** [ID]*

Institute of Imaging and Computer Vision, RWTH Aachen University, Aachen, Germany

* fdennis.eschweiler@lfb.rwth-aachen.de (DE); johannes.stegmaier@lfb.rwth-aachen.de (JS)

## Abstract

Automated image processing approaches are indispensable for many biomedical experiments and help to cope with the increasing amount of microscopy image data in a fast and reproducible way. Especially state-of-the-art deep learning-based approaches most often require large amounts of annotated training data to produce accurate and generalist outputs, but they are often compromised by the general lack of those annotated data sets. In this work, we propose how conditional generative adversarial networks can be utilized to generate realistic image data for 3D fluorescence microscopy from annotation masks of 3D cellular structures. In combination with mask simulation approaches, we demonstrate the generation of fully-annotated 3D microscopy data sets that we make publicly available for training or benchmarking. An additional positional conditioning of the cellular structures enables the reconstruction of position-dependent intensity characteristics and allows to generate image data of different quality levels. A patch-wise working principle and a subsequent full-size reassemble strategy is used to generate image data of arbitrary size and different organisms. We present this as a proof-of-concept for the automated generation of fully-annotated training data sets requiring only a minimum of manual interaction to alleviate the need of manual annotations.

## 1 Introduction

Current developments of fluorescence microscopy imaging techniques allow to acquire vast amounts of image data, capturing different cellular structures for various kinds of biological experiments in 3D and over time [1, 2]. The growing amount of data that needs to be analyzed, demands an automation of image processing tasks [3]. To this end, automated approaches became widely used and especially machine learning and deep learning-based approaches offer powerful tools for detection, segmentation, and tracking of different cellular structures [4, 5]. However, learning-based approaches need large annotated data sets to become robust and generalist. On the other hand, the creation of manually annotated data sets is very time-consuming and tedious, causing those data sets to be rarely available. Although there are many classical and machine learning-assisted annotation tools accessible [6–9], which reduce the annotation time for biological experts, especially the dense annotation of 3D data remains difficult. Current

**Data Availability Statement:** The code framework used in this publication is available at https://github.com/stegmaierj/CellSynthesis and archived at https://doi.org/10.5281/zenodo.5118755. The synthesized 3D microscopy image data sets are

available at https://doi.org/10.17605/OSF.IO/
E6N7B, https://doi.org/10.17605/OSF.IO/9RG2D
and https://doi.org/10.17605/OSF.IO/5EFM9.

**Funding:** This work was funded by the German
Research Foundation DFG (DE, Grant No STE
2802/2-1). The funders had no role in study
design, data collection and analysis, decision to
publish, or preparation of the manuscript.

**Competing interests:** The authors have declared
that no competing interests exist.

approaches propose to reduce annotation efforts and increase generalizability of machine learning-based approaches by collecting a manifold of annotated image data from slightly different domains, creating a highly diverse training data set [10]. This supports the claim that a large amount of annotated training data is a key factor in creating robust approaches.

To reduce the need of manual annotations, simulation tools have been proposed, which rely on known imaging and microscopy parameters [11–13]. Those techniques can be used to automatically create annotated data samples, but they rely on precise prior knowledge about each specific experimental setup. With the recent success of generative adversarial networks (GANs) [14] and their progressive developments, simulation approaches became more realistic, less parameter dependent and, thus, more applicable. Techniques have been proposed to directly synthesize 2D cellular structures [15–17] or simulate 3D cellular structures [18], which in turn can be used to synthesize realistic 3D images with existing classical approaches [19] or GAN-based approaches [20].

In this paper we demonstrate how a conditional generative adversarial network [21] can be used to transform large realistic 3D annotation masks of multi-cellular structures into realistic 3D microscopy image data in a patch-based manner, by employing a patch merging strategy to obtain seamless results. Furthermore, the proposed conditioning leverages the synthesis of different image quality levels to further increase diversity of simulated image data. We introduce different approaches for the simulation of these 3D annotation masks, which can be chosen with respect to the complexity of cellular structures and the availability of manually obtained annotation masks. As a contribution to the community, two fully-annotated 3D synthetic data sets are made publicly available to potentially serve as training or benchmark data sets for 3D detection and segmentation approaches for fluorescently labeled nuclei and membranes.

## 2 Generation of cellular 3D structures

The generation of realistic annotation masks of cellular structures is crucial for the synthesis of realistic image data, since unrealistic and overly artificial structures can impede structural correlation between annotation masks and synthetic images, as demonstrated in [15]. If features of cellular structures in the annotation masks $m \in \mathcal{M}$ deviate too much from those in the image data $i_{real} \in \mathcal{I}_{real}$, the generator network $G$ can no longer find a suitable mapping and starts to deform annotated structures to resemble those in the image domain, which would cause the annotated structures to no longer match the structures in the synthetic image data. Therefore, it is crucial to use simulation techniques that reproduce structures as realistically as possible.

Realistic annotation masks can be obtained in several ways, depending on the availability of suitable segmentation approaches, time for manual expert annotations and complexity of cellular structures. Besides manual annotation or automated segmentation, we experimented with a few modelling approaches that can be used for the simulation of both organism shape and cell structures. The discussed simulation is divided into three parts (Fig 1): first, the generation of the foreground region outlining an organism shape, second, determining positions of cellular structures within the foreground region and finally, modelling of cellular structures at each given position. Ultimately, cells are either represented as single nuclei or as cellular membranes, *i.e.*, as a structural mesh within the foreground region.

### 2.1 Geometrical modelling

Modelling cellular structures with geometrical functions is the least complex simulation technique discussed in this paper and, consequently, offers a limited set of shapes that can be represented. Nevertheless, it offers a straightforward possibility to model cellular structures.

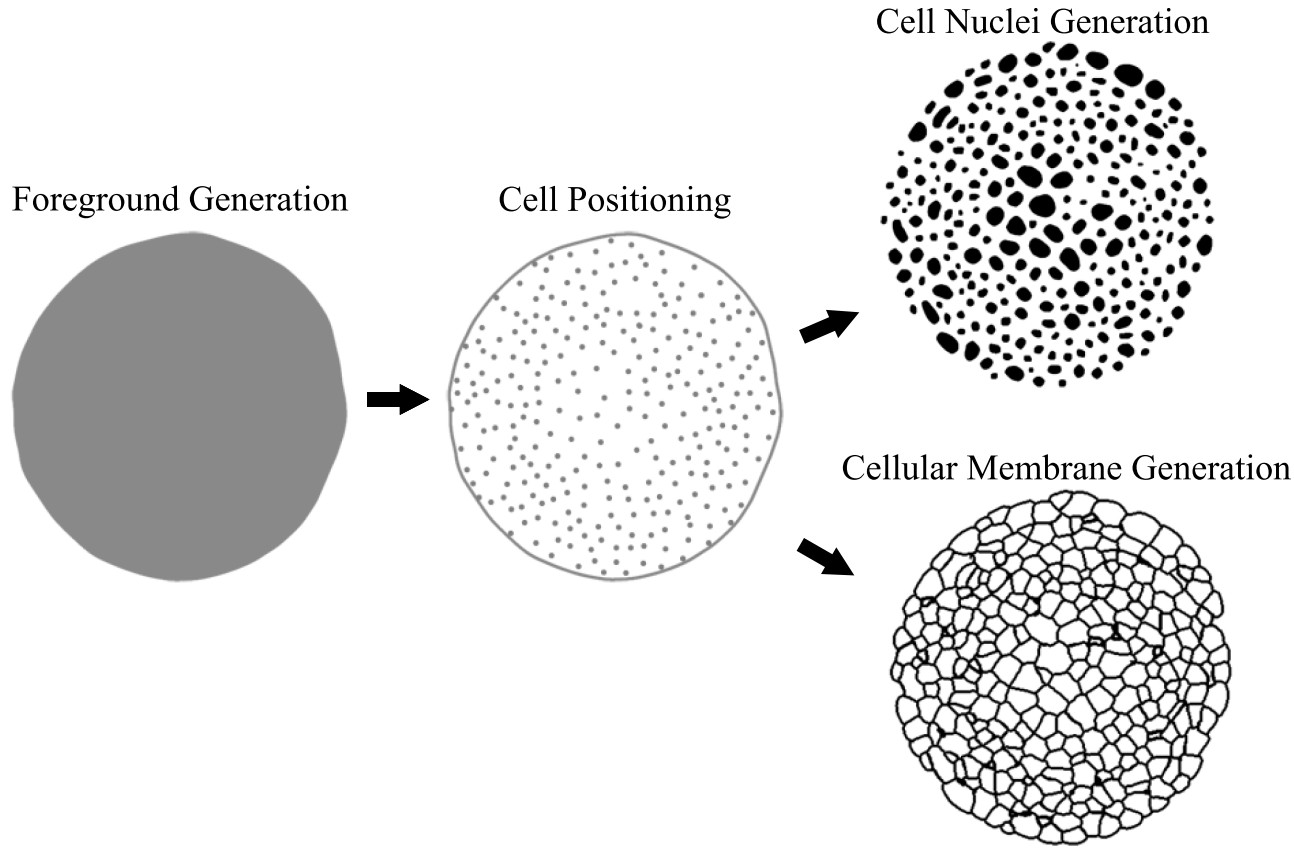

**Fig 1. Qualitative 2D illustration of the 3D mask simulation pipeline.** The simulation process generates an annotation mask in three refinement steps, starting with organism shape generation, adding cell positions within the foreground region and final nuclei or membranes structure generation at each respective positions.

For cell positioning we follow a layer-wise assembly, starting to add layers of cells at the outer boundary of the organism and we progressively add additional layers inwards, if reasonable. The additional layer locations are determined by shrinking the organism shape and randomly placing new cells along the determined boundary, with random small offsets to add further realism. Cell distribution and density in each layer and layer thickness are adjusted using prior knowledge about cell sizes to avoid unnatural shape variations in the 3D space. For the simulation of cellular membranes, each foreground voxel $\mathbf{x}_{fg}$ is assigned to the closest cell position $\mathbf{x}_{cell}$, creating a Voronoi-like tessellation of the foreground region. Since the resulting segments show unnaturally straight edges, an additional weighting is applied to the distance calculation, which is adapted from [22] and bends the planes between different segments, resulting in the distance metric

$$dist(\mathbf{x}_{fg}, \mathbf{x}_{cell,j}) = \frac{\|\mathbf{x}_{fg} - \mathbf{x}_{cell,j}\|_2}{\gamma_j} \qquad (1)$$

with weight $\gamma_j$ being fixed for cell position $\mathbf{x}_{cell,j}$. The final tessellation is used to either create annotation masks for cell membranes by considering planes between segments or instance segmentations of whole cells by considering entire segments.

## 2.2 Statistical shape models

If a set of annotations is already available, characteristic shape parameters can be extracted and used to generate additional data. Statistical shape models offer a way to determine distinctive shape modes and encode them in an accessible way [23] to generate shapes by changing only a desirable small number of parameters. As a prerequisite, each voxelized shape needs to be sampled at predefined angular rays starting at the organism centroid or cell centroid and it needs to be converted into a list of 3D boundary coordinates $\mathbf{p} = (x_1, y_1, z_1, \ldots, x_n, y_n, z_n)^T$. From coordinate lists of all shapes, the mean shape $\bar{\mathbf{p}} = \frac{1}{k}\sum_{j=1}^{k} \mathbf{p}_j$ and the covariance matrix $\Sigma_{\mathbf{p}} = \frac{1}{k-1}\sum_{j=1}^{k}(\mathbf{p}_j - \bar{\mathbf{p}})(\mathbf{p}_j - \bar{\mathbf{p}})^T$ are computed. To identify distinctive modes, an eigenvalue decomposition of the covariance matrix $\Sigma_{\mathbf{p}}$ is computed, resulting in eigenvectors $\phi$ and corresponding eigenvalues $\lambda$, while modes are ordered so that $\lambda_1 \geq \lambda_2 \geq \ldots \geq \lambda_N$. Accordingly, the first eigenvectors encode most of the shape variance and account for the most distinctive modes. New shapes $\mathbf{p}_{new}$ are generated by altering the mean shape by a linear combination of the first $n$ modes, while $n \leq N$ can be chosen as a required level of detail:

$$\mathbf{p}_{new} = \bar{\mathbf{p}} + \sum_{j=1}^{n} \mathbf{b}_j \phi_j. \tag{2}$$

Influence of each mode is defined by a weight vector $\mathbf{b}$, which is randomly sampled from but restricted to the limit $b_j \in [-3\lambda_j, 3\lambda_j]$ to only allow the generation of shapes that lie within the determined variance range. The obtained list of boundary points is transformed to a voxelized mask representation using a Delaunay triangulation. Possible shape variations learned from a public data set [24] are shown in Fig 2.

## 2.3 Spherical harmonics

Spherical harmonics (SH) offer another way of encoding shapes in a compact format [25], facilitating the generation of new shapes by changing a small number of parameters. This concept is related to the Fourier transform, but is inherently suitable for spherical shapes. The generation of shapes with spherical harmonics is accomplished by defining each shape as a composition of a predefined set of $R$ weighted orthonormal basis functions, which can be identified by an order $l$ describing the level of high frequency shape components and a degree $m$ describing a particular variant of an order. The resulting set of functions is organized in a pyramid scheme with $l \geq 0$ and $-l \leq m \leq l$ and each spherical harmonics basis function can be computed by

$$Y_j = Y_l^m(\theta, \phi) = \sqrt{\frac{2l+1}{4\pi} \cdot \frac{(l-m)!}{(l+m)!}} \cdot P_l^m(\cos\theta)e^{i \cdot m \cdot \phi} \tag{3}$$

with $P_l^m(\cos\theta)$ describing the Legendre polynomials of degree $m$ and order $l$ and parameters $\theta$ and $\phi$ denoting the spherical angular sampling coordinates. An individual 3D shape $S_{SH}$ is defined as a weighted linear combination of the spherical harmonic basis functions

$$S_{SH} = \sum_{j=1}^{R} c_j \cdot Y_j, \tag{4}$$

with weight $c_j$ specifying the influence of the corresponding spherical harmonic basis functions $Y_j$. Since the first basis function ($l = m = 0$) already represents a perfect sphere scalable by a single parameter $c_1$, spherical harmonics are well-suited to describe and model cellular structures [26–28]. Note that SH representations are limited to star-convex shapes, which we, however,

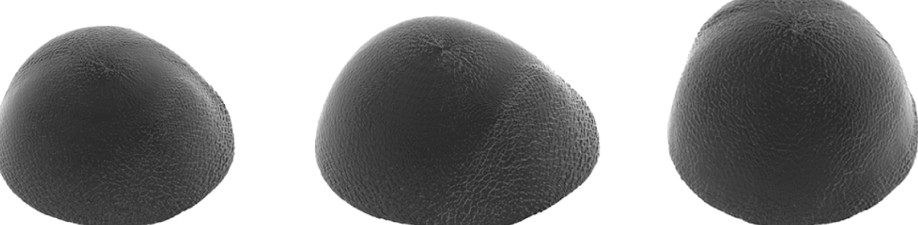

**Fig 2. Overview of possible spherical shapes initialized with statistical shape models.** Different shape variations are visualized, with statistics learned from a public data set [24].

found sufficient for simulation of nuclei and various early stage embryo shapes. Since the set of basis functions is already formulated, the generation of shapes reduces to choosing the number $R$ of desired spherical harmonics and determining corresponding weights $c_{1,\ldots,R}$. $R$ is chosen by specifying the desired level of detail, *i.e.*, the order of high frequency shape components $l$, since $R = (l + 1)^2$ when using all available degrees $m$. Weights $c_{1,\ldots,R}$ are designed to have an exponentially decreasing magnitude to approximate smooth shapes, while further diversity is added by initializing each weight by a random value drawn from a standard normal distribution $\mathcal{N}(0, 1)$. This results in

$$c_l^m = r \cdot w(l, m) \cdot e^{-\gamma m}, \ c_0^0 = r \tag{5}$$

with $r$ defining the approximate radius of the spherical shape, $w(l, m)$ constituting the random initialization assigned to the coefficient for degree $m$ and order $l$ and $\gamma$ controlling the smoothness of the resulting shape. Examples for different $\gamma$ are shown in Fig 3.

## 3 Synthesis of 3D microscopy data

The proposed synthesis pipeline is designed to transform binary annotations of cellular structures in 3D image data into realistic microscopy image data. Generating realistic image data from annotation masks offers several advantages, since it allows to control the outline and geometry of structures that should be generated and it allows to generate image data sets that inherently include corresponding annotations, independent from image quality or degradation.

For the synthesis of realistic 3D microscopy image data, a generative adversarial network (GAN) [21] is used, which allows for an unsupervised translation between mask annotations $\mathcal{M}$ and microscopy image data $\mathcal{I}$. This concept includes two individual networks (Fig 4): a generative network $G$, which is trained to learn a mapping $G(m)$ from a mask $m \in \mathcal{M}$ into an

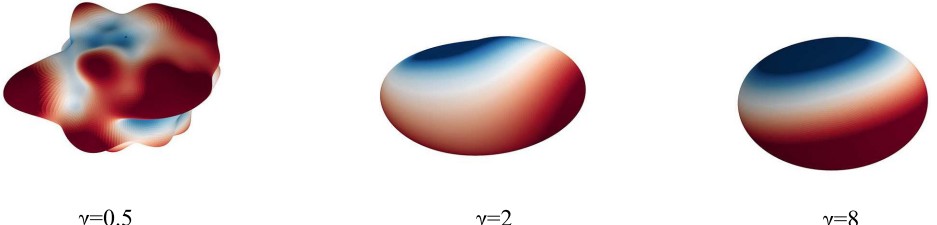

| $\gamma$=0.5 | $\gamma$=2 | $\gamma$=8 |

**Fig 3. Overview of possible spherical shapes initialized with spherical harmonics.** Shapes are shown for different values of $\gamma$ and colors indicate a positive (red) or negative (blue) deviation from a perfect sphere.

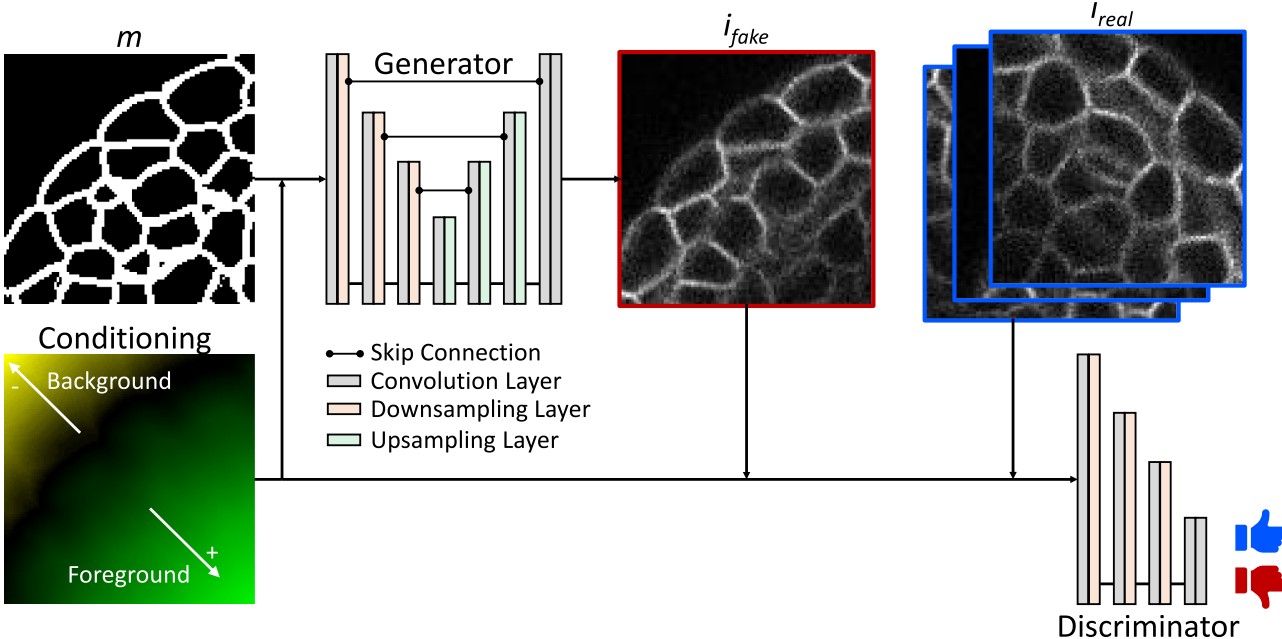

**Fig 4. Schematic of the conditional generative adversarial network.** The generator transforms annotation masks $m$ into realistic images $i_{fake} \in \mathcal{I}_{fake}$, which are assessed by the discriminator using realistic image data $i_{real} \in \mathcal{I}_{real}$. The positional conditioning is used to generate and assess positional image characteristics. Note that, for simplicity, visualizations are provided in 2D, despite processing being done in 3D.

image $i_{fake} \in \mathcal{I}_{fake}$, and a discriminator network $D$, which is trained to distinguish between real $i_{real} \in \mathcal{I}_{real}$ and generated $i_{fake} \in \mathcal{I}_{fake}$ image data. Consequently, the training procedure is an adversarial assembly, where the generator wants to outperform the discriminator by generating realistic images, while the discriminator aims to be as good as possible in distinguishing real from fake images. In principle, the generator is trained to simulate the image formation process of a microscope without the necessity of prior knowledge about the acquisition process parameters.

For the generator network a 3D U-Net [29] is used and the pixel-shuffle technique [30] is incorporated for higher quality upsampling in the decoding path. The generator training objective consists of two separate losses, including the adversarial loss $\mathcal{L}_{adv}$ as formulated in [31] to assess the generated image quality and an identity loss $\mathcal{L}_{identity} = \|i_{real} - G(i_{real})\|_1$ to support the generation of realistic intensity characteristics and impose a structural correlation between the input and output. A Patch-GAN is used as discriminator network and trained with adversarial losses for real and fake images, as proposed in [31].

Due to large data sizes of 3D microscopy image data and memory limitations of current graphics processing units (GPUs), processing 3D images as a whole is not feasible, demanding a patch-based processing. Processing patches, however, causes two sources of errors, including tiling artifacts when reassembling the full-size image and loss of global positional information for structures within each patch, which is very crucial in generating realistic image degradation and positional intensity characteristics. Firstly, tiling artifacts are avoided by overlapping neighbouring patches by a fixed margin $d_{overlap}$ and averaging the overlap regions based on a concentric weight map, decreasing the influence of voxels based on their distance to the patch center, as done in [26]. Additionally, to avoid potential patch border artifacts, output patches are cropped by a fixed margin $d_{crop}$ before full-size assembly. Secondly, global positional

information of patches are maintained by providing an additional conditioning as input, encoding the position of each voxel in relation to the specimen's boundary, which is motivated by the occurrence of depth-dependent intensity decay in microscopy imaging. We choose a hyperbolic tangent function to further indicate if a voxel lies within a specimen (foreground) or in a background region to enable the generation of more detailed noise features. To consider different influences of distances within the foreground and background regions, respectively, the positional function is adjustable for both regions individually, leading to the following formulation:

$$f_{pos}(\mathbf{x}) = \begin{cases} tanh(\quad dist(\mathbf{x})/\alpha) & \text{if } \mathbf{x} \text{ in foreground} \\ tanh(-dist(\mathbf{x})/\beta) & \text{if } \mathbf{x} \text{ in background} \end{cases} \tag{6}$$

where $\mathbf{x}$ is the 3D position vector of a voxel and $dist(\mathbf{x})$ measures the smallest distance to the specimen's boundary. Scaling parameters $\alpha$ and $\beta$ are used to adjust the saturation of the hyperbolic tangent to realistically simulate the limited penetration depth of the imaging system. Applying this function to each voxel, an encoded distance map is created, which matches the spatial dimension of the corresponding annotation mask and is used as conditional input for both, the generator and discriminator network (Fig 4).

During training with small data sets, we observed the generator to preferably generate specific intensity and noise patterns, resulting in a slight overfitting to the training samples. In order to counteract this problem, augmentation strategies could be employed to further increase diversity of the image training data, which, however, needs to be configured carefully to create as much diversity as possible, but to include as little alterations as possible to still enable a smooth and realistic generator training. We follow the concept proposed in [32], stating that overfitting to small data sets can be avoided by utilizing an adaptive discriminator augmentation (ADA), *i.e.*, augmentation of real $\mathcal{I}_{real}$ and generated $\mathcal{I}_{fake}$ images before they are being assessed by the discriminator. This helps to include as much diversity as possible by simultaneously controlling the impact of those alterations. To establish this control, a set of augmentations is predefined and each individual transformation is sequentially applied to an image with probability $p_{aug}$. To automatically identify the optimal value for $p_{aug}$, an overfitting measure is determined, which calculates the fraction of real image samples $\mathcal{I}_{real}$ that are correctly classified by the discriminator, leading to the following heuristic:

$$r_{ADA} = \mathbb{E}[sign(D(\mathcal{I}_{real}))] \tag{7}$$

As proposed in [32], a target value of $r_{ADA} = 0.6$ is chosen, indicating that the augmentation probability $p_{aug}$ should increase for larger values and decrease for smaller values by a step size of $\delta_{aug}$ every $e_{aug}$ epochs. To enhance robustness of the training procedure, we use rectified Adam [33] as an optimizer for all networks.

## 3.1 Putting it all together

Our data generation pipeline consists of two major parts (Fig 5). First, the simulation or acquisition of annotations of cellular structures, which either represent nuclei or cellular membranes. The generation of annotations has different stages, starting with organism shape generation, cell positioning and final generation of cellular structures, while each simulation stage can be implemented by manual interaction or automated detection and segmentation approaches. Second, the final annotations are used to generate image data with the proposed GAN training and they are further used to generate corresponding instance segmentations, allowing to create fully-annotated data sets for further utilization.

Acquisition/Simulation of Annotations

Image Synthesis

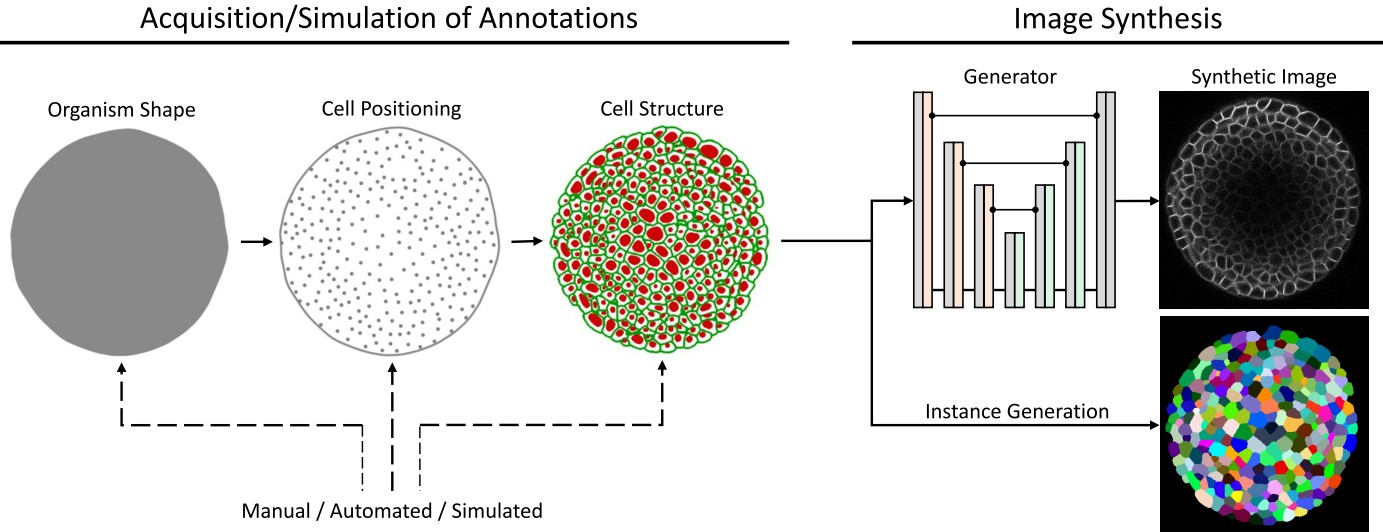

**Fig 5. Qualitative 2D overview of our 3D data generation pipeline.** It comprises the acquisition or simulation of annotations and the image synthesis. Annotations can be obtained from manual, automated, and simulation approaches, and final cellular annotations are used to generate corresponding instance segmentations.

## 4 Experiments and quality assessment

Two publicly available data sets are used for the evaluation of our approach:

**Data set** $\mathcal{D}_{AT}$ The first data set includes 125 3D image stacks showing fluorescently labeled cell membranes in six different *Arabidopsis thaliana* shoot apical meristem that were imaged with confocal microscopy at a resolution of $0.22 \times 0.22 \times 0.26\ \mu m^3$ per voxel using a 63×/1.0 N. A. water immersion objective [24]. Additionally, 3D instance segmentations are available for all image stacks, which were automatically obtained and partly manually corrected by the authors. All images have a spatial size ranging between $326 \times 367 \times 107$ and $512 \times 512 \times 396$ voxel.

**Data set** $\mathcal{D}_{DR}$ The second data set consists of a total of 394 3D image stacks showing fluorescently labeled cell membranes and nuclei in two different *Danio rerio* (DR1 and DR2) [34]. Image stacks were captured with multi-photon microscopy at a resolution of $1.37 \times 1.37 \times 1.37$ $\mu m^3$ per voxel [34]. No ground truth annotations are available and all images have a spatial size of $512 \times 512 \times z$ voxel, while $z$ is ranging from 104 to 120.

Ideally, a sufficiently large fully-annotated data set is available for the training of the proposed image synthesis pipeline, as cellular structures have to be as similar as possible to the structures in the real data. However, due to the lack of fully-annotated 3D image data sets, a more realistic scenario would consider automatically-annotated image stacks or desirably even fully-simulated annotations, where no tedious and time-consuming manual annotation is required. To evaluate different aspects of our simulation pipeline and influences of different annotation acquisition techniques, we conduct multiple experiments ranging from an ideal case scenario to more practical scenarios, as described in the following.

For all experiments in this section real image data is used as the target domain and annotation masks from different sources are used as source domain, *i.e.*, as input to the generative adversarial network. Training is performed for 5000 epochs using a patch size of $128 \times 128 \times 64$ voxel and distance scalings of $\alpha = \beta = 100$ (Eq 6). During each epoch only one randomly

located patch is extracted from each of the images to increase data diversity during training. Furthermore, the set of random augmentations available for the adaptive discriminator augmentation includes linear intensity scaling in the range [0.6, 1.2], additive Gaussian noise sampled from $\mathcal{N}(0, 0.1)$, voxel shuffling in a randomly located region of size $25 \times 25 \times 25$ voxel, randomly located inpaintings of size $15 \times 15 \times 15$ voxel and linear intensity reduction along a random dimension. Augmentation updates are performed every $e_{aug} = 1$ epochs with a step size of $\delta_{aug} = 0.05$ to allow adjusting the augmentation parameters in a reasonable number of epochs. For resembling of the full-size image, an overlap and crop of $d_{overlap} = d_{crop} = (30, 30, 15)$ are used.

## 4.1 Image synthesis with manually annotated data $\mathcal{S}_{\text{manual}}$

As a first experiment, the 3D instance segmentation masks of the $\mathcal{D}_{AT}$ data set are used as input for the generative adversarial network, which serves as a baseline to assess the capability of the proposed synthesis pipeline. The six different specimens of this data set are divided into training and test sets, using plants 1, 2, 4 and 13 for training and plants 15 and 18 for testing.

Qualitative results are illustrated in Fig 6 (left column pair), comparing different views of real and synthetic image data. As further qualitative measures the intensity profile for a real and the corresponding synthetic center xz- and yz-slice is generated, integrated over the z dimension and plotted along the x and y dimension, respectively (Fig 6, $Membrane_{manual}$). Additionally, intensity spectra are created for a xy-slice of the real and synthetic image (Fig 6, $Membrane_{manual}$). Both illustrations demonstrate the similarity between real and synthetic image data in the spatial and in the spectral domain. A large portion of the images contains noise, which is non-deterministic and can vary between images without impairing the image quality. Furthermore, especially since the proposed concept operates in an unsupervised fashion, the trained networks tend to optimize towards a consensus of intensity patterns within the training data set, causing the resulting images to contain less prominent or differently located intensity peaks by still realistically reconstructing the given structures. These aspects cause a visual discrepancy between high-frequency areas in the spectra and peaks in the intensity profiles.

As quantitative quality metrics the normalized root mean square error (NRMSE), structural similarity index measure (SSIM) and the zero mean normalized cross-correlation (ZNCC) are used (Table 1, $Membrane_{manual}$). By considering that a large portion of the image is background or random noise and only a small portion of the image contain cellular structures, these scores prove the synthetic data to be realistic, supporting the impression from the qualitative assessment.

## 4.2 Image synthesis with automatically obtained data $\mathcal{S}_{\text{automatic}}$

In a second experiment, annotations of cellular structures are automatically obtained for the $\mathcal{D}_{DR}$ data set. For cellular membranes, we used the approach described in [35], which is a 3D extension of the Cellpose approach [10] using a 3D convolutional neural network to represent cell shapes as generalist gradient maps. An iterative post-processing technique is used to reconstruct the individual cell instances. Training of the neural network is performed for 1000 epochs using the manually annotated $\mathcal{D}_{AT}$ training data set and the model is applied to images of the $\mathcal{D}_{DR}$ image data. For cell nuclei we used the approach proposed in [36], which makes use of a Laplacian of Gaussian (LoG) scale-space responding to a predefined range of cell sizes. The shape-sensitive output is used to determine cell locations and to facilitate a watershed-

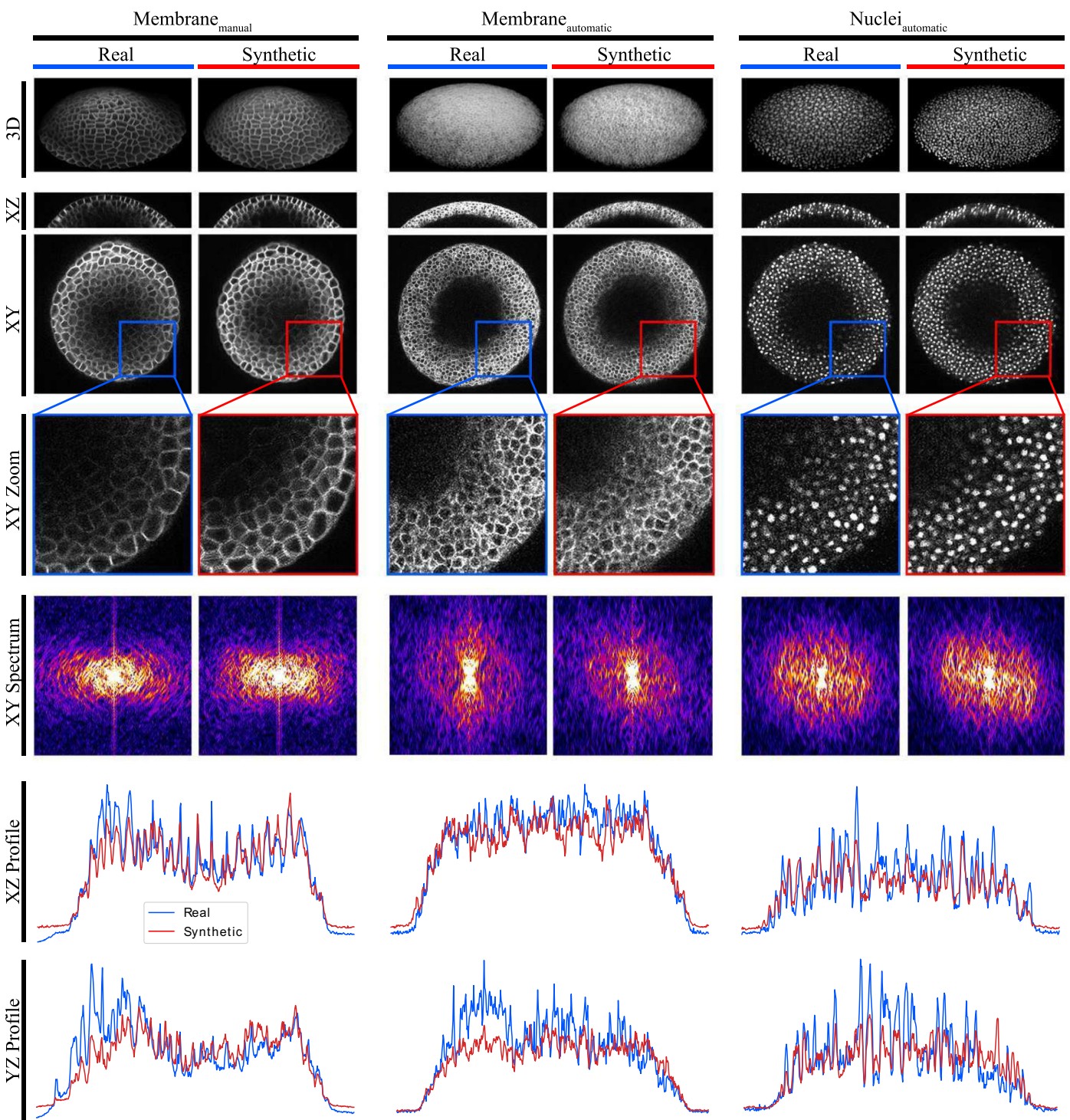

**Fig 6. Synthetic image data.** Different views of real image data (blue columns) in comparison to synthetic image data (red columns) generated by our GAN approach. Examples are shown for different experiments using manually corrected masks from the $\mathcal{D}_{AT}$ data set [24] (*Membrane_manual*), and automatically annotated masks for membranes (*Membrane_automatic*) and nuclei (*Nuclei_automatic*) from the $\mathcal{D}_{DR}$ data set [34]. Additionally, the spectra and intensity profiles of different slices are shown as qualitative metrics.

**Table 1. Quantitative assessment of image quality.** Quality scores obtained for the different synthetic data sets.

|  | NRMSE | SSIM | ZNCC |
|---|---|---|---|
| $Membrane_{manual}$ | 0.130 | 0.658 | 0.739 |
| $Membrane_{automatic}$ | 0.123 | 0.584 | 0.847 |
| $Nuclei_{automatic}$ | 0.119 | 0.617 | 0.751 |
| $Nuclei_{simulated}$ | 0.129 | 0.706 | 0.632 |

based segmentation of cell nuclei. Parameters for this segmentation approach are manually tweaked to produce good results.

Since there is no ground truth segmentation data available for the $\mathcal{D}_{DR}$ data set, segmentation approaches serve as a straightforward way to create annotations that represent the shapes and distribution of cellular structures as realistically as possible. Note that no further post-processing or manual corrections are performed to represent a realistic scenario with minimum manual interaction. The first specimen of the $\mathcal{D}_{DR}$ data set (DR1) was used for training of the GAN and the second specimen (DR2) was used for testing.

Qualitative results for the synthetic membrane data and further qualitative measures, including the intensity profile of center xz- and yz-slices and intensity spectra of xy-slices are illustrated in Fig 6 ($Membrane_{automatic}$). Results for the synthetic nuclei data are illustrated in Fig 6 ($Nuclei_{automatic}$).

By keeping in mind, that the precision of the automatically obtained annotation masks can not be assessed, again NRMSE, SSIM and ZNCC are used as quantitative quality metrics (Table 1, $Membrane_{automatic}$ and $Nuclei_{automatic}$), which are similar to the scores obtained in the previous experiment $\mathcal{S}_{manual}$. Qualitative and quantitative measures and figures again demonstrate success in generation of realistic 3D microscopy image data from unrefined, automatically generated annotation masks.

## 4.3 Image synthesis on simulated data

For training of deep learning-based synthesis approaches it is important to use annotations of cellular structures that are as similar as possible to structures in the real image data to prevent the generation of image artifacts and structural misalignment [15]. In this experiment, we further assessed if the application of a trained model requires the same level of structural correlation. To this end, we simulated annotation masks for both, cellular membranes and cell nuclei from very sparse annotations and used the models from $\mathcal{S}_{manual}$ and $\mathcal{S}_{automatic}$ to synthesize real microscopy image data.

As a first scenario, only automatically obtained point annotations from the $\mathcal{D}_{DR}$ data set as described in $\mathcal{S}_{automatic}$ are considered and, subsequently, cell nuclei are modelled at each position utilizing spherical harmonics. Spherical harmonics coefficients are randomly constructed using a smoothness factor $\gamma = 5$ and approximate radii $r \in \mathcal{N}(9, 1)$ pixel. For each of the 200 raw image stacks of the test set (DR2), one corresponding mask is simulated and the synthesized image data is generated using the trained nuclei model from $\mathcal{S}_{automatic}$ (Fig 7, top). Since positional correspondences are maintained and only structural characteristics differ between real and synthetic image data, again NRMSE, SSIM and ZNCC are used as quantitative quality metrics (Table 1, $Nuclei_{simulated}$). The obtained scores are similar to the scores obtained from the previous experiments and allow us to again conclude success in the generation of realistic 3D image data.

As a second scenario, statistical shape models are utilized to generate organism shapes similar to the specimen of the $\mathcal{D}_{AT}$ data set and, subsequently, cellular structures are simulated

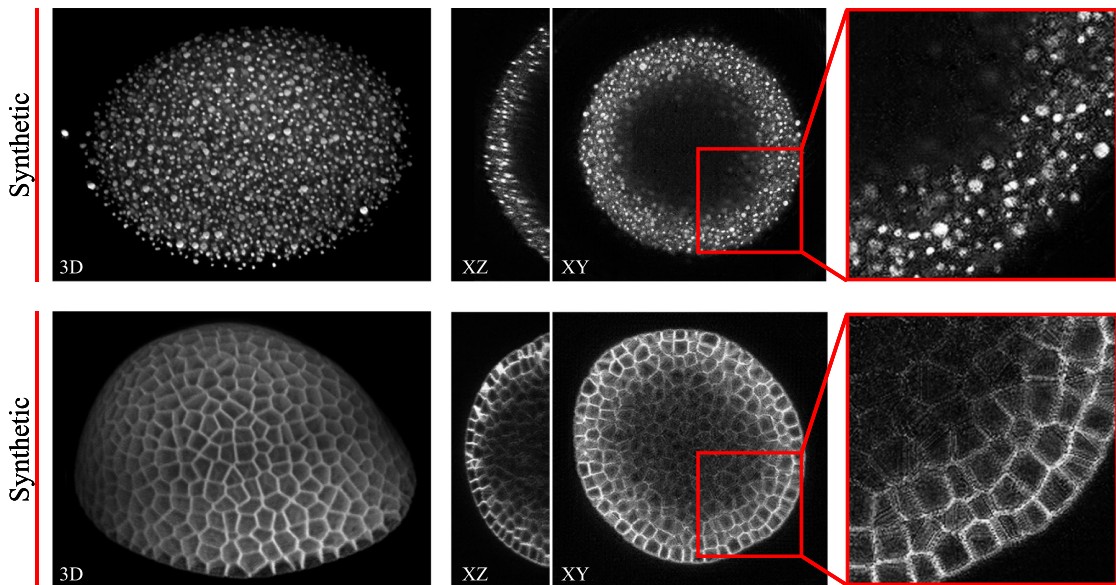

**Fig 7. Fully synthetic image data.** Different views of synthetic nuclei image data (top) and synthetic membrane image data (bottom) generated by our GAN approach using the simulated annotation masks.

within the foreground region by using geometrical modelling. Corresponding synthesized image data is obtained with the trained model from $\mathcal{S}_{manual}$ (Fig 7, bottom). Accurate shape model statistics are derived from the foreground segmentation of the corresponding manually obtained annotation masks, although foreground segmentation has also been shown to be automatable, *e.g.*, by automated 3D interpolation techniques using very sparse manually drawn 2D outlines [37]. This scenario renders pixel-level metrics to be no longer feasible due to the missing direct structural correspondences between real and synthetic image samples. However, since the data is ultimately generated to alleviate the need of manually annotated data for learning-based approaches, we assess how accurate a 3D segmentation approach [38] performs on the $\mathcal{D}_{AT}$ test data set, when trained on synthetic image data. This multi-class segmentation approach predicts probability maps of background, membrane structures, and centroids, which helps to demonstrate how the segmentation of different structures is affected by the image synthesis. Finally, a watershed-based post-processing technique is used to reconstruct individual cell instances. Therefore, 125 simulated masks are generated and split into 82 for training and 43 for testing to match the quantity of image data from the real data set (Fig 7, right). Subsequently, the segmentation approach is trained on the synthetic data for 1000 epochs using a patch size of $128 \times 128 \times 64$ voxel and it is applied to both, the test data of the synthetic data set (Syn2Syn) and the test data of the $\mathcal{D}_{AT}$ data set (Syn2Real), including plants 15 and 18. Additionally, the segmentation approach is trained on the train split of the original $\mathcal{D}_{AT}$ data set and applied to the real (Real2Real) and the synthetic (Real2Syn) test split. Furthermore, a mixed data set is constructed by combining both real and synthetic data sets to further increase data diversity. The model trained on the combined training set is tested on real (Mix2Real) and synthetic (Mix2Syn) test data. The probability map of each class, including background, cellular membrane and cell centroids, is thresholded by an individual value $t_{background}$ = 0.1, $t_{membrane}$ = 0.4 and $t_{centroid}$ = 0.2 and, afterwards, an average intersection over union (IoU) is computed as a metric for each of the three classes (Fig 8, top). To obtain representative

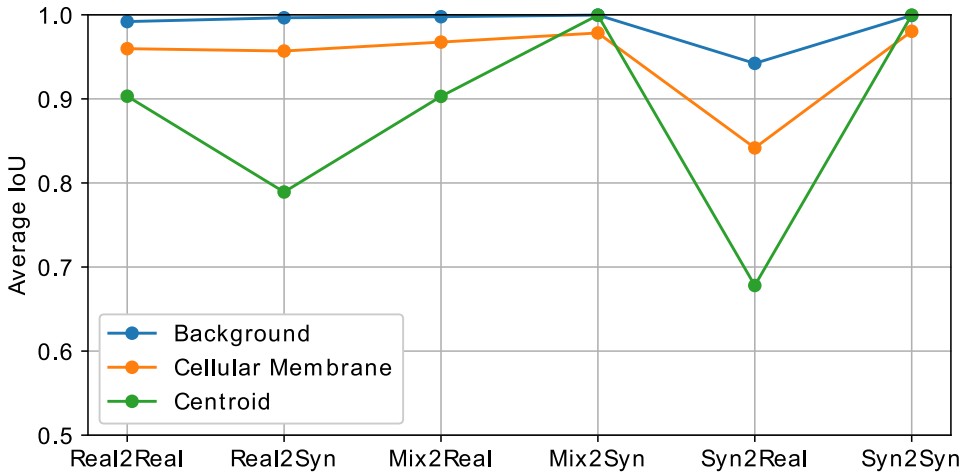

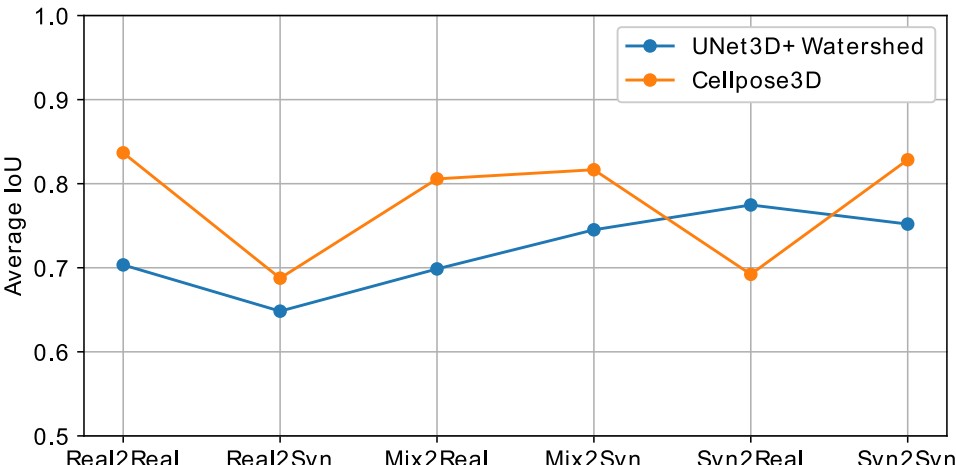

**Fig 8. Segmentation scores obtained for different data setups.** Multi-class segmentation scores obtained with the approach from [38] (top) and instance segmentation scores obtained with the approaches from [38, 35] (bottom) trained on real data (Real2Real, Real2Syn) and synthetic data (Syn2Real, Syn2Syn) and a mixed data set containing real and synthetic data (Mix2Real, Mix2Syn). Trained models are applied to real and synthetic data, respectively.

results, we allow predictions to be within a small range around the ground truth masks, which accommodates the small sizes of cellular structures and centroids. The allowed misplacement distances are 1 voxel for the background and cellular membrane classes and 5 voxel for the centroid class. Additionally, instance segmentations are obtained from the multi-class approach and another robust instance segmentation approach [35] and resulting average intersection over union (IoU) scores are plotted in Fig 8 (bottom). Overall high segmentation scores support that the generated data is realistic. However, due to differences in structure and intensity characteristics, the transfer training setups (Real2Syn and Syn2Real) show a decrease in segmentation scores, which is more severe when training on synthetic and testing on real data. Since centroids represent the class with the smallest object sizes, the highest difference in obtained segmentation scores are observed for this class. Nevertheless, the obtained segmentation scores when training on synthetic and testing on real data motivate the utilization of

synthetic microscopy image data as a training data set for segmentation approaches, especially considering that the synthetic training data can be obtained with a minimum of manual interaction. Mixed training setups (Mix2Real and Mix2Syn) further motivate the incorporation of synthetic image data to diversify the training data set and improve results. Due to the controlled generation of synthetic image data, the resulting fully-annotated data sets don't contain large segmentation inaccuracies, which accounts for the score difference between the pure training cases (Real2Real and Syn2Syn).

## 4.4 Image synthesis on different quality levels

The proposed positional conditioning of the generative adversarial network allows to control the quality level of the generated data by altering the positional foreground parameter $\alpha$ (Eq 6). This property is investigated by using different scalings $\alpha \in \{10, 50, 100, 500, 1000\}$ to generate different quality levels of the $\mathcal{D}_{AT}$ test set (plants 15 and 18) using the generative network from experiment $\mathcal{S}_{manual}$. Image slices of three different quality levels are depicted in Fig 9 (top),

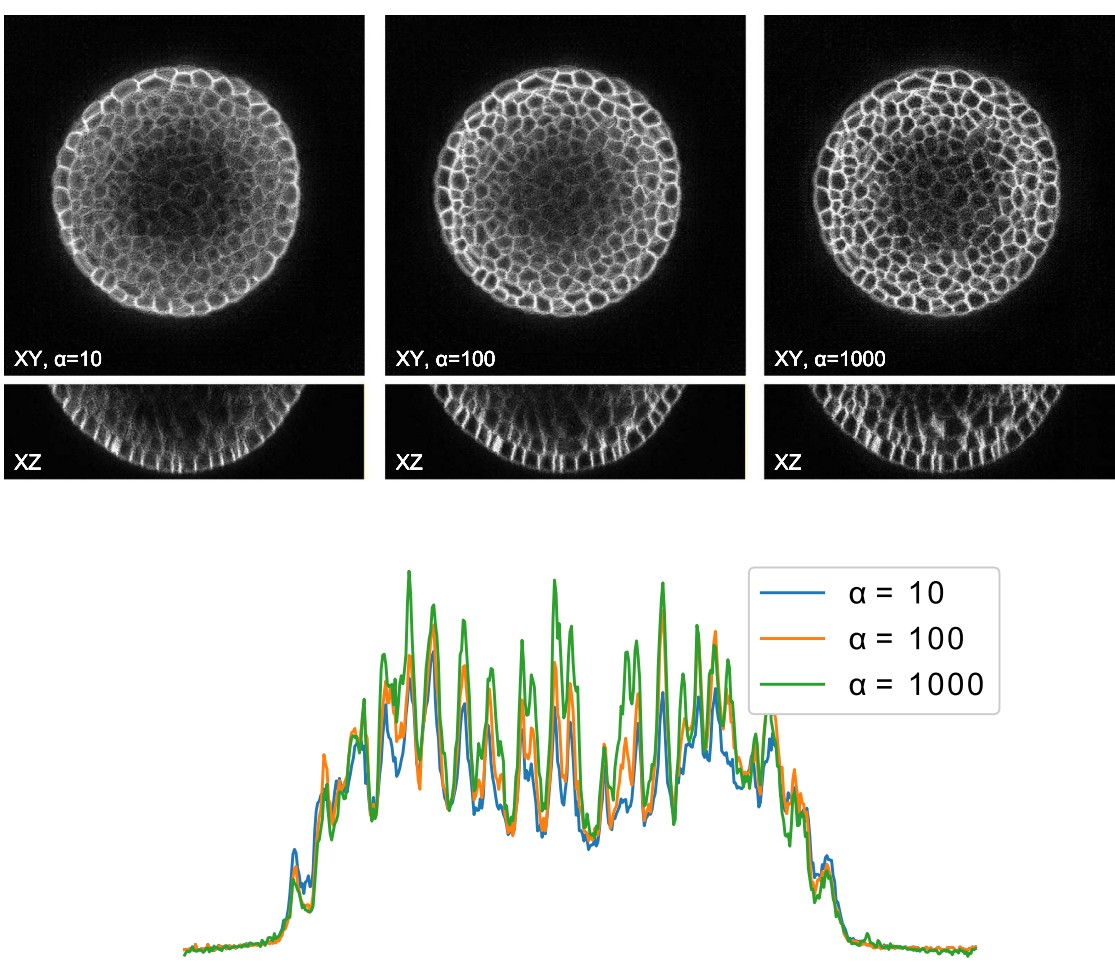

**Fig 9. Qualitative assessment of altered image quality.** 2D slices of 3D synthetic image data generated by our GAN approach using the same manually corrected mask from [24], with different foreground distance scalings $\alpha$ (top). For the center xz-slice, intensities are integrated over the z dimension and plotted along the x dimension (bottom).

**Table 2. Quantitative assessment of altered image quality.** Quality scores obtained for synthetic image data generated on different quality levels by varying $\alpha$ from Eq 6, including the normalized root mean square error (NRMSE), structural similarity index measure (SSIM), zero mean normalized cross-correlation (ZNCC) and peak signal-to-noise ratio (PSNR).

| | $\alpha = 10$ | $\alpha = 50$ | $\alpha = 100$ | $\alpha = 500$ | $\alpha = 1000$ |
|---|---|---|---|---|---|
| NRMSE | 0.134 | 0.130 | 0.130 | 0.136 | 0.140 |
| SSIM | 0.640 | 0.654 | 0.658 | 0.660 | 0.643 |
| ZNCC | 0.704 | 0.735 | 0.739 | 0.721 | 0.703 |
| PSNR | 17.485 | 17.802 | 17.768 | 17.376 | 17.079 |

alongside the integrated intensity profile of the center xz-slices (bottom), with both of these qualitative illustrations showing the decaying intensity for poor quality levels.

To evaluate the quality of the generated data, we make use of the structural correspondence to the real image data and compute NRMSE, SSIM, ZNCC and PSNR as metrics (Table 2). The obtained scores slightly deteriorate for both, worse and improved image quality, which is caused by the changes in structural quality and noise content. Since these changes are to be expected and the scores remain in the same value range, the claim of generating realistic image data holds.

Additionally, we obtain multi-class segmentations for the real and the synthetic test image data on all quality levels using the approach proposed in [38] and consult these accuracies as a further quality metric. Again, class probability maps are thresholded by an individual value $t_{background} = 0.1$, $t_{membrane} = 0.4$ and $t_{centroid} = 0.2$ and, afterwards, an aggregated intersection over union (IoU) is computed as a metric for each of the three classes (Fig 10, top). Due to the small sizes of membrane structures and centroids, small displacements would have a large, hardly interpretable impact on the obtained segmentation accuracies and, therefore, both classes are allowed to lie within a small distance to the ground truth of 1 voxel for membrane structures and 5 voxel for centroids. Instance segmentation results for both segmentation approaches [35, 38] are evaluated by computing average intersections over union (IoU) scores (Fig 10, bottom).

The baseline (Fig 10, dashed line) is computed from segmentation scores obtained on the real $\mathcal{D}_{AT}$ test set. Segmentation scores for cellular membranes increase with increasing image quality until nearly reaching the baseline score, demonstrating the progressively increasing quality of cellular structures. Background accuracy decreases with increasing image quality, which is caused by the diminishing noise content within the foreground region, causing the segmentation approach to confuse regions within cells with background regions. Additionally, we observed harsh and unnatural intensity transitions for the highest quality level $\alpha = 1000$, which causes further segmentation inaccuracies and points out an upper boundary for a natural increase of the image quality. Although these facts produce segmentation artifacts, overall background segmentation scores stay close to the baseline scores and promote the claim of generating increasing image quality. This problem also affects the centroid segmentation and results in decreased segmentation accuracy for the best quality level. Due to the small sizes of centroid segmentations, accuracy scores are very sensitive to small imprecisions and, thus, cause a discrepancy to the baseline scores. The same score progression is shown for instance segmentations obtained with two segmentation approaches, with a minimum decrease of around 0.1 IoU from the baseline results. Nevertheless, the increasing progression of scores according to the increasing quality level demonstrates the utility of the synthesized image data.

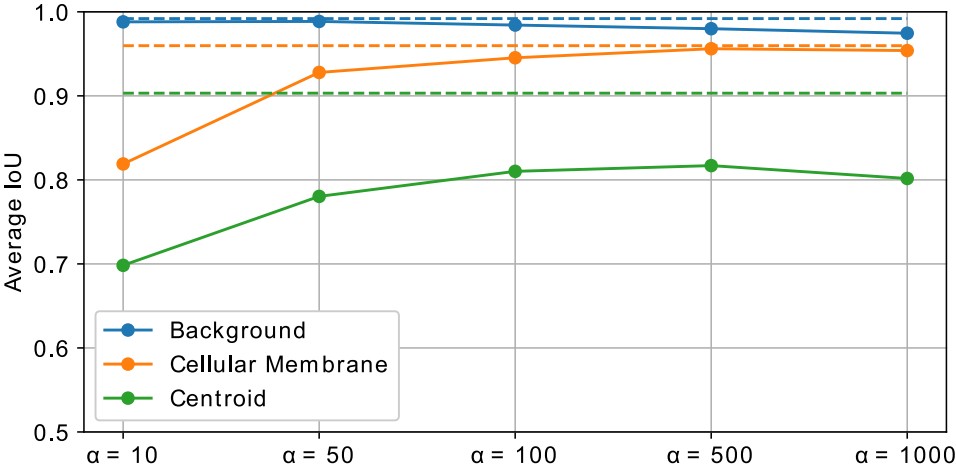

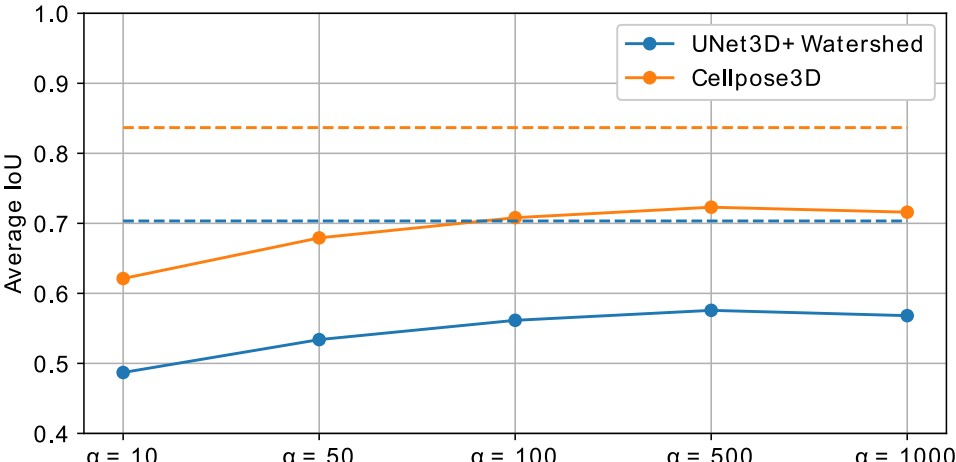

**Fig 10. Segmentation scores obtained on altered image quality.** Multi-class segmentation scores [15] (top) and instance segmentation scores [15, 35] (bottom) obtained for synthetic image data generated on different quality levels by varying $\alpha$ from Eq 6. Dashed lines represent results obtained on the original test data.

## 5 Conclusion and availability

In this work we demonstrated how a conditional generative adversarial network can be utilized to synthesize realistic 3D fluorescence microscopy image data from binary annotations of cellular structures in a patch-based manner, allowing for a controlled generation of fully-annotated 3D image data sets. Due to the patch-based approach, data of arbitrary size can be generated, enabling an image-based simulation of entire organisms. Different techniques for the generation of annotations of cellular structures were shown in several experiments with varied necessity of manual interaction. The conducted experiments demonstrated the generation of realistic 3D image data from an ideal scenario with manually obtained annotations being available, to a practical scenario with annotations simulated from very sparse detections or segmentations. Using predefined cellular structures as a source domain allows to generate fully-annotated data sets, which, we believe, leverages the usage of learning-based approaches

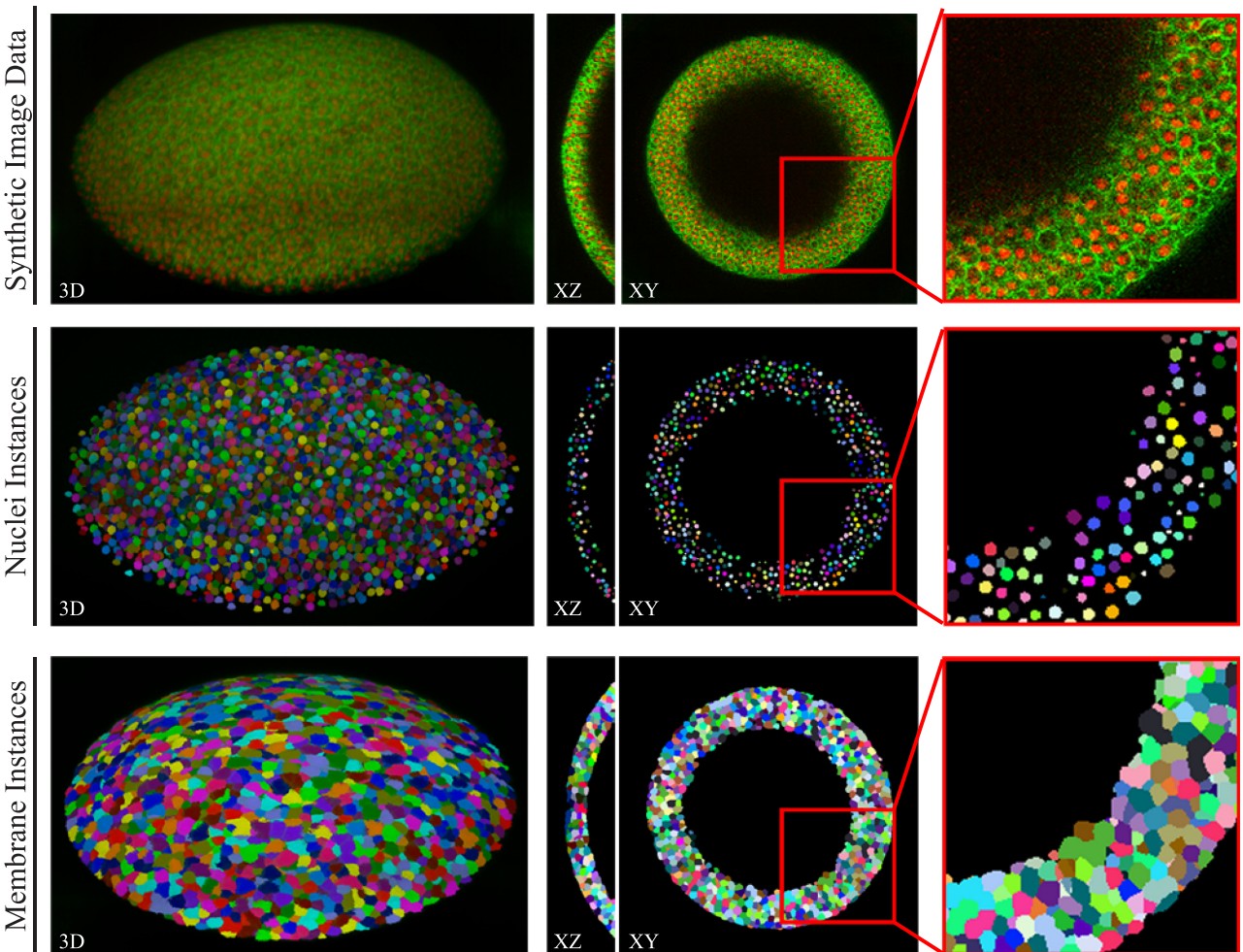

**Fig 11. Public two channel data set.** Samples of the first published data set, including generated 3D image data of nuclei and cellular membrane (top) and the corresponding automatically obtained instance segmentations for nuclei (middle) and membranes (bottom).

for many biomedical experiments. The proposed conditioning of the generative adversarial network establishes the ability to generate data on different quality levels, which further strengthens control over the generated image data for increased data diversity or detailed benchmarking. In future work we plan to extend our experiments to additional organisms, striving for a larger collection of publicly available fully-annotated 3D microscopy data sets and we plan to further extend the simulation to time series data.

In order to make the generated data accessible to the community for training, we publish two data sets comprising nuclei (DOI: 10.17605/OSF.IO/9RG2D) and cellular membrane (DOI: 10.17605/OSF.IO/5EFM9) data generated from automatically obtained annotations $\mathcal{S}_{automatic}$ with the corresponding instance segmentation data (Fig 11). Furthermore, we publish three different quality levels of the $\mathcal{D}_{AT}$ test set (plants 15 and 18, Fig 12) with instance segmentations being available from [24], to provide image data with increasing quality for potential benchmarkings. We decided to publish synthetic image data scaled with $\alpha \in 10, 100, 500$ (DOI: 10.17605/OSF.IO/E6N7B), as those levels proved to be of increasingly better quality for segmentation approaches (Fig 10). We also publish the code repository used to generate 3D

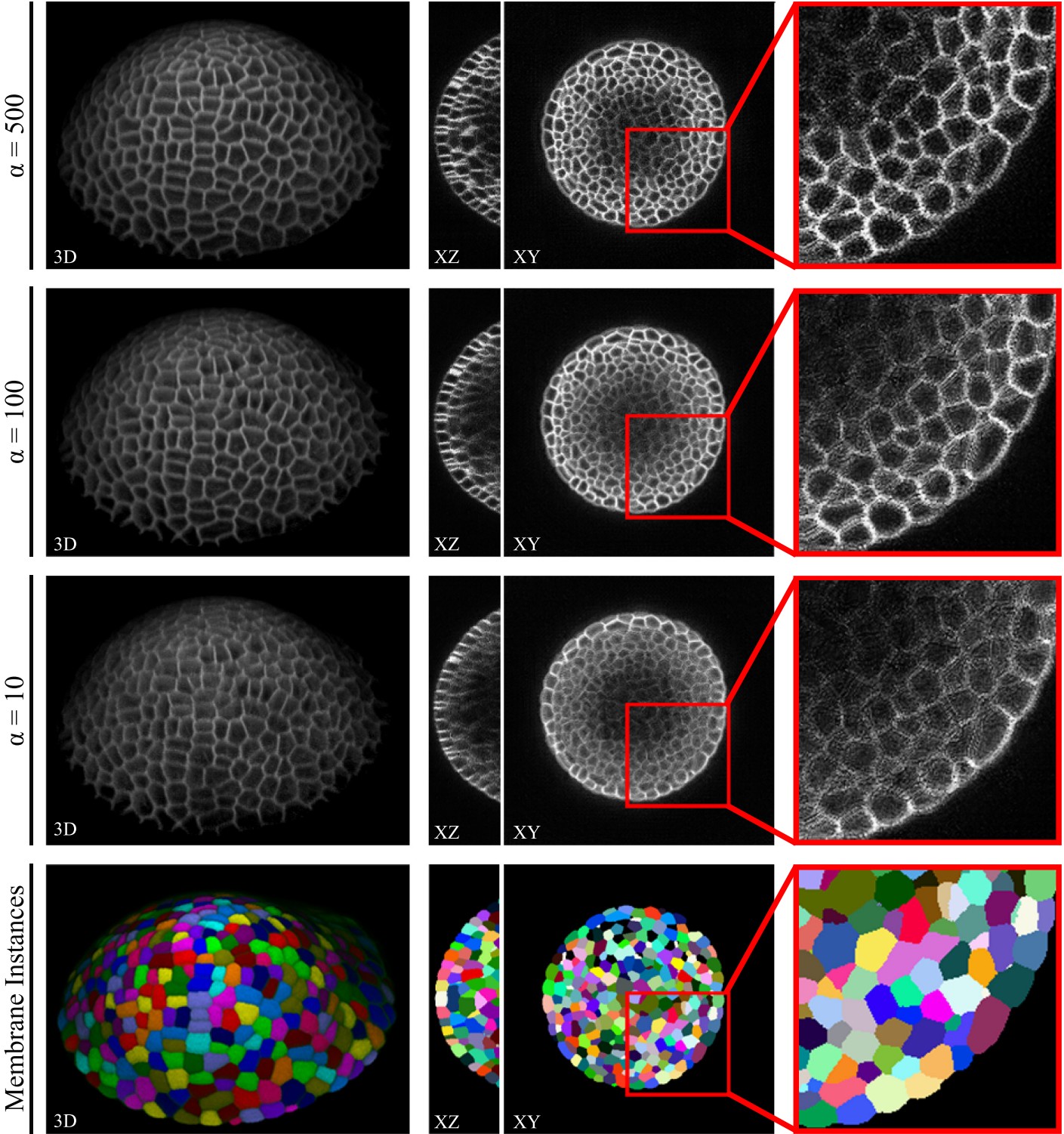

**Fig 12. Public altered quality data set.** Samples of the second published data set, including 3 different quality levels of the same cellular membrane structures. Corresponding instance segmentations (bottom) are available from [24].

annotations and synthetic image data, which is available at https://github.com/stegmaierj/CellSynthesis (DOI: 10.5281/zenodo.5118755).

## Author Contributions

**Conceptualization:** Dennis Eschweiler.

**Data curation:** Dennis Eschweiler.

**Formal analysis:** Dennis Eschweiler.

**Funding acquisition:** Johannes Stegmaier.

**Investigation:** Dennis Eschweiler.

**Methodology:** Dennis Eschweiler, Malte Rethwisch.

**Project administration:** Johannes Stegmaier.

**Resources:** Johannes Stegmaier.

**Software:** Dennis Eschweiler, Malte Rethwisch, Simon Koppers.

**Supervision:** Johannes Stegmaier.

**Validation:** Dennis Eschweiler.

**Visualization:** Dennis Eschweiler, Mareike Jarchow.

**Writing – original draft:** Dennis Eschweiler.

**Writing – review & editing:** Dennis Eschweiler, Johannes Stegmaier.

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
