## [Decision Letter · Decision Letter 0]

1 Oct 2021

PONE-D-21-255013D fluorescence microscopy data synthesis for segmentation and benchmarkingPLOS ONE

Dear Dr. Eschweiler,

Thank you for submitting your manuscript to PLOS ONE. After careful consideration, we feel that it has merit but does not fully meet PLOS ONE’s publication criteria as it currently stands. Therefore, we invite you to submit a revised version of the manuscript that addresses the points raised during the review process.

 Please consider the revisions suggested by the reviewers.

We look forward to receiving your revised manuscript.

Kind regards,

Kristen C. Maitland, Ph.D.

Academic Editor

PLOS ONE

Journal Requirements:

Reviewers' comments:

Reviewer's Responses to Questions

**Comments to the Author**

1. Is the manuscript technically sound, and do the data support the conclusions?

Reviewer #1: Yes

Reviewer #2: Yes

2. Has the statistical analysis been performed appropriately and rigorously? 

Reviewer #1: Yes

Reviewer #2: Yes

3. Have the authors made all data underlying the findings in their manuscript fully available?

Reviewer #1: Yes

Reviewer #2: Yes

4. Is the manuscript presented in an intelligible fashion and written in standard English?

Reviewer #1: Yes

Reviewer #2: No

5. Review Comments to the Author

Reviewer #1: The authors present work addressing the need for automated generation of training data sets for deep-learning based segmentation of cell membranes and nuclei. Different methods for generating the synthetic training sets are evaluated qualitatively and quantitatively, as well as how effective they are when used to train a segmentation network that is applied to real data. This type of methods analysis is important for biologists to appreciate and understand the nuance and challenges still existing with deep-learning segmentation tools. It also highlights the importance of data quality and provides a way to asses optimal data quality for future acquisitions. The annotated data sets made available may be very useful to research groups that are still in the process of manually annotating their own data and don't have access to a useful already trained network.

I recommend this work be published with a few suggestions for the author's consideration:

My biggest suggestion is to consolidate the figures. I think they could be combined for a total of 10 figures at most that tell the story in a more succinct way.

Line 5: the comma after "analyzed" should be removed

Line 7: a comma could be included after "segmentation"

When discussing spherical harmonics, it may be useful for a general audience to compare to Fourier Transform, a more familiar weighted sum of basis functions. Though, the presentation given is already done well.

Line 202: The "in conclusion" seems unnecessary

Line 209: It would be good to mention the accuracy measures that will be used here.

Figure 5 legend: I would add commas so the text reads "Annotations can be obtained from manual, automated, and simulation approaches, and final cellular annotations are used to...."

Figure 4: If there are these visual examples provided of the spherical harmonic models, it would be nice to see the statistical shape models also.

Line 216: It would be useful to mention the imaging lens numerical aperture and physical pixel sizes here from ref 32. for quick understanding of the spatial resolution of the data sets.

Line 222: It would be useful to mention this data set was acquired with a multi-photon fluorescence microscope (ref 33).

Line 250: By integrating the z-dimension for the intensity profiles you lose the ability to asses the depth-dependent performance of the GAN. It would be interesting to have at least two -perhaps an "upper" and "lower" profile if not a variety of individual xy planes at different z positions.

Line 252: The similarity between the real and synthetic xz intensity spectra could be qualified with the visually obvious discrepancies that happen in the high-frequency areas. What accounts for those discrepancies could be discussed.

Line 262 and 263: If a few words describing the approaches in [34] and [35] could be used here it would improve the readability.

Line 304: "allow to again conclude the generation of realistic 3D image data" is missing a subject, consider "allow us to again conclude..."

Line 316: Please describe the approach in [37] for the audience's quick reference.

Line 358: It would be interesting to also quantify the different "quality levels" of the synthesized data sets in terms of signal-to-noise ratio, contrast ratio, or some other normal image quality metric.

Reviewer #2: The paper is overall well presented but there are a couple confusing aspects to the presentation.

There is no top down explanation of the logic of the various experimental mixes of images and masks which makes things more challenging than necessary to understand

There are a few grammatically problematic sentences e.g. "The obtained scores are similar to the scores obtained from the previous experiments and allow [us] to again conclude [success in] the generation of realistic 3D image data."

It would be helpful to situate this work with respect to other CNN based training data simulation efforts e.g. NucleAIzer Hollandi et al Cell Systems 2020

6. PLOS authors have the option to publish the peer review history of their article (what does this mean?). If published, this will include your full peer review and any attached files.

Reviewer #1: No

Reviewer #2: **Yes: **Anthony Santella

---

## [Author Response · Author response to Decision Letter 0]

3 Nov 2021

We thank you for considering our manuscript for publication as an article in PLOS ONE and we are excited to revise our manuscript based on the very helpful comments and suggestions from both reviewers.

As one of the major changes made to our manuscript, we swapped sections 2 and 3 for improved readability and better coherence of the whole story. Further changes and detailed responses to the reviewers’ suggestions are listed in the following. We are looking forward to your assessment of the improvements made to the manuscript and hope all requests have been fulfilled.

Reviewer #1: 

My biggest suggestion is to consolidate the figures. I think they could be combined for a total of 10 figures at most that tell the story in a more succinct way.

- We totally agree that there have been to many figures impairing the reader’s experience. The total number of figures has been reduced by combining or removing previous figures, although we ended up at a quantity slightly above 10. 

Line 5: the comma after "analyzed" should be removed

- Removed the comma, thanks for pointing this out. 

Line 7: a comma could be included after "segmentation" 

- Added the comma, thanks for the hint.

When discussing spherical harmonics, it may be useful for a general audience to compare to Fou-rier Transform, a more familiar weighted sum of basis functions. Though, the presentation given is already done well.

- We agree that the concept of the Fourier transform is very likely more familiar to the reader, and we added a hint to the similarities between both concepts to the spherical harmonic section. Thanks for pointing this out. However, we feel that a full comparison between both approaches would lead to an overly complex and lengthy explanation, that might lead to confusion, as Fourier transform is not used later on.

Line 202: The "in conclusion" seems unnecessary

- This phrase was removed. 

Line 209: It would be good to mention the accuracy measures that will be used here.

- This should refer to the general quality of the generated structures. However, we think that this sentence is rather confusing and does not provide reasonable information, which is why we re-moved it. 

Figure 5 legend: I would add commas so the text reads "Annotations can be obtained from man-ual, automated, and simulation approaches, and final cellular annotations are used to...."

- Commas were added to the legend. 

Figure 4: If there are these visual examples provided of the spherical harmonic models, it would be nice to see the statistical shape models also.

- For consistency we added a few examples of shapes generated with statistical shape models. 

Line 216: It would be useful to mention the imaging lens numerical aperture and physical pixel sizes here from ref 32. for quick understanding of the spatial resolution of the data sets.

- Good point, we missed this information entirely. The missing information was added to the data set description.

Line 222: It would be useful to mention this data set was acquired with a multi-photon fluores-cence microscope (ref 33).

- Microscopy technique and imaging resolution have been added to the data set description. 

Line 250: By integrating the z-dimension for the intensity profiles you lose the ability to assess the depth-dependent performance of the GAN. It would be interesting to have at least two -perhaps an “upper” and “lower” profile if not a variety of individual xy planes at different z positions.

- Assessing the depth-dependent performance was the intention of these plots. To get a better overview of the distribution of intensities, we added further plots showing the distribution over YZ. To our understanding, these plots help to assess how well the cumulative intensity for differ-ent thicknesses of the specimen can be reconstructed. We think, XY profiles would be harder to interpret and would only show depth-dependent performance when considering quite a few z po-sitions, which would start to be cluttering. If you have any tips on how we could represent this in-formation in a more compact format, we would be happy to extend the figure. 

Line 252: The similarity between the real and synthetic xz intensity spectra could be qualified with the visually obvious discrepancies that happen in the high-frequency areas. What accounts for those discrepancies could be discussed.

- One of our possible interpretations is that those discrepancies are caused by non-deterministic components like noise, and differences in signal intensities. We added further interpretations and explanations to Section 4.

Line 262 and 263: If a few words describing the approaches in [34] and [35] could be used here it would improve the readability.

- A short description of both approaches has been added, which hopefully helps to get a better un-derstanding of the experimental setup. 

Line 304: “allow to again conclude the generation of realistic 3D image data” is missing a subject, consider “allow us to again conclude…”

- Thanks for pointing this out, we corrected the sentence. 

Line 316: Please describe the approach in [37] for the audience’s quick reference.

- We added further explanations of the approach’s concept for a quick reference. 

Line 358: It would be interesting to also quantify the different “quality levels” of the synthesized data sets in terms of signal-to-noise ratio, contrast ratio, or some other normal image quality metric.

- The PSNR was added as further metric. Due to the different value ranges of PSNR and the other metrics, we replaced the plot with a table.

Reviewer #2:

There is no top down explanation of the logic of the various experimental mixes of images and masks which makes things more challenging than necessary to understand.

- A general explanation of the choice of our experiments was indeed missing and would help to clar-ify the whole evaluation section. A few sentences explaining our intention of the experiments was added to the beginning of Section 4.

There are a few grammatically problematic sentences e.g. "The obtained scores are similar to the scores obtained from the previous experiments and allow [us] to again conclude [success in] the generation of realistic 3D image data."

- Thanks for pointing this out, we corrected the sentence. 

It would be helpful to situate this work with respect to other CNN based training data simulation efforts e.g. NucleAIzer Hollandi et al Cell Systems 2020

- We think that a direct comparison to this approach would not be straightforwardly possible, as the generation of images in 2D and patch-based 3D faces significantly different challenges. However, we agree that referring to this approach would complement the state-of-the-art literature, which is why we added a citation of this approach to the introduction section.

---

## [Editor Report · Decision Letter 1]

11 Nov 2021

3D fluorescence microscopy data synthesis for segmentation and benchmarking

PONE-D-21-25501R1

Dear Dr. Eschweiler,

We’re pleased to inform you that your manuscript has been judged scientifically suitable for publication and will be formally accepted for publication once it meets all outstanding technical requirements.

Kind regards,

Kristen C. Maitland, Ph.D.

Academic Editor

PLOS ONE
---

## [Editor Report · Acceptance letter]

23 Nov 2021

PONE-D-21-25501R1 

3D fluorescence microscopy data synthesis for segmentation and benchmarking 

Dear Dr. Eschweiler:

I'm pleased to inform you that your manuscript has been deemed suitable for publication in PLOS ONE. Congratulations! Your manuscript is now with our production department. 

Kind regards, 

on behalf of

Dr. Kristen C. Maitland 

Academic Editor

PLOS ONE